# Photoprotective Effect of Dietary Galacto-Oligosaccharide (GOS) in Hairless Mice via Regulation of the MAPK Signaling Pathway

**DOI:** 10.3390/molecules25071679

**Published:** 2020-04-06

**Authors:** Min Geun Suh, Gi Yeon Bae, Kyungae Jo, Jin Man Kim, Ki-Bae Hong, Hyung Joo Suh

**Affiliations:** 1Department of Integrated Biomedical and Life Science, Graduate School, Korea University, Seoul 02841, Korea; adcdr565@naver.com (M.G.S.); rldus530@naver.com (G.Y.B.); 2BK21Plus, College of Health Science, Korea University, Seoul 02841, Korea; kyungae11@korea.ac.kr; 3Department of Food Marketing and Safety, Konkuk University, Seoul 05029, Korea; jinmkim@konkuk.ac.kr

**Keywords:** galacto-oligosaccharide, MAPK, MMP, TEWL

## Abstract

This study investigated the suppression of photoaging by galacto-oligosaccharide (GOS) ingestion following exposure to ultraviolet (UV) radiation. To investigate its photoprotective effects, GOS along with collagen tripeptide (CTP) as a positive control was orally administered to hairless mice under UVB exposure for 8 weeks. The water holding capacity, transepidermal water loss (TEWL), and wrinkle parameters were measured. Additionally, quantitative reverse-transcription polymerase chain reaction and Western blotting were used to determine mRNA expression and protein levels, respectively. The GOS or CTP orally-administered group showed a decreased water holding capacity and increased TEWL compared to those of the control group, which was exposed to UVB (CON) only. In addition, the wrinkle area and mean wrinkle length in the GOS and CTP groups significantly decreased. Skin aging-related genes, matrix metalloproteinase, had significantly different expression levels in the CTP and GOS groups. Additionally, the tissue inhibitor of metalloproteinases and collagen type I gene expression in the CTP and GOS groups significantly increased. Oral administration of GOS and CTP significantly lowered the tissue cytokine (interleukin-6 and -12, and tumor necrosis factor-α) levels. There was a significant difference in UVB-induced phosphorylation of JNK, p38, and ERK between the GOS group and the CON group. Our findings indicate that GOS intake can suppress skin damage caused by UV light and has a UV photoprotective effect.

## 1. Introduction

Skin is one of the main barriers of the body, protecting it from external stimuli [1]. Like in other barriers, various microorganisms live in the skin and are reported to behave differently depending on skin status, age, and external environment [2]. Changes in skin microorganisms are related to skin diseases; however, their precise effects on disease-related protection, recovery, or aggravation are not clear yet. Improvements in skin microbial flora have been reported to benefit skin health [3]. It has been reported that skin diseases, such as atopic dermatitis, can be treated by changing the intestinal microflora through the consumption of probiotics and prebiotics. Unlike probiotics, which are used for the propagation of certain microorganisms, prebiotics are optional ingredients that act as nutrient sources for intestinal microorganisms, resulting in health benefits due to various intestinal microbial metabolic changes [4,5].

Prebiotics can be described as dietary fiber because they are indigestible; however, not all fibers are prebiotics, because they are not processed by intestinal microorganisms, which are selective [5,6]. The most widely used prebiotics are fructo-oligosaccharides (FOS), inulin, and galacto-oligosaccharides (GOS). GOS are potent non-digestible prebiotics composed of three to 10 molecules of galactose and glucose, and the elongation products of β-galactosidase. GOS molecules are known to contain a varying degree of β-glycosidic linkages and promote digestive and immune health through the growth of beneficial bacteria [7]. In our previous reports, GOS prebiotic activity has been reported to contribute to skin health, both alone and as a component of complex probiotics, by enhancing the growth of beneficial intestinal microbial flora [8]. GOS administration in hairless mice inhibited the increase in transepidermal water loss (TEWL) and increased the water holding capacity; moreover, it is reported to upregulate the CD44 gene, which is closely related to the maintenance of hyaluronic acid homeostasis. In addition, it was reported that after an intake of 1 g GOS, the water holding capacity increased and transdermal moisture loss significantly changed [9]. GOS as a prebiotic in an atopic dermatitis-induced NC/Nga mouse model minimized the production of interleukin 10 and suppressed the production of cytokines, such as interleukin 17 [10].

Although many studies have demonstrated that GOS intake has a moisturizing and anti-inflammatory effect, no elaborate studies on the mechanism of action in GOS have been conducted. Therefore, this study aimed to investigate the effect and mechanism of skin photoprotection by GOS administration.

## 2. Results and Discussion

### 2.1. Effect of GOS Intake on Water Holding Capacity and TEWL in UVB-Irradiated Hairless Mice

When the skin is repeatedly exposed to ultraviolet rays, reactive oxygen species (ROS) are generated and promote the production of proinflammatory cytokines; this activates various signaling systems and increases the inflammatory response [11,12]. The increased expression of matrix metalloproteinases (MMPs) also increases collagen degradation. These changes in skin tissue increase skin thickness and alter the structure of the dermal layer, damaging the normal skin barrier function that regulates the water holding capacity and TEWL. Maintenance of the skin barrier is a dynamic process that involves the coordination of various signaling systems, supplementation of the missing lipids, and induction of biosynthesis [13].

In the ultraviolet (UV) B control (CON; irradiated and untreated mice) group, the water holding capacity and TEWL significantly changed compared to those in the normal (NOR; unirradiated mice, Figure 1A: *p* < 0.05, Figure 1B: *p* < 0.001) group. In contrast, the water holding capacity of both the GOS intake at 200 mg/kg (GOS; irradiated and GOS-treated mice) and collagen tripeptide intake at 200 mg/kg (collagen tripeptide (CTP); irradiated and CTP-treated mice) groups was significantly higher than that of the CON (Figure 1A CTP and GOS: *p* < 0.05). No significant difference was noted in the NOR. TEWL was also significantly lower in the GOS and CTP groups than in the CON (Figure 1B, CTP and GOS: *p* < 0.05). The oral intake of GOS has an effect similar to that of collagen intake, which has already been shown to have a positive influence on the recovery of the skin barrier function [14].

### 2.2. Effect of GOS Intake on Wrinkle Area and Mean Wrinkle Length in UVB-Irradiated Hairless Mice

UVB can penetrate the skin up to the basal layer to produce harmful ROS, causing inflammation that promotes skin aging [15]. Oxidative stress plays a synergistic role in skin damage caused by UV. ROS are induced by UV, and their accumulation eventually causes inflammation and wrinkle formation in the skin [16].

To analyze the effect of GOS ingestion on skin wrinkle formation induced by UV, wrinkles on the dorsal skin of hairless mice were analyzed after 8 weeks of UV irradiation treatment. The wrinkle area and mean wrinkle length significantly increased in the CON compared to those in the NOR (Figure 2A,B: *p* < 0.001). In addition to UVB treatment, treatment with GOS and CTP significantly lowered the wrinkle area (CTP and GOS: *p* < 0.001) and mean wrinkle length (CTP: *p* < 0.01) compared to those in the CON. Similar to the intake of the positive control CTP, UVB irradiation after the intake of GOS showed anti-wrinkle effects.

### 2.3. Effect of GOS Intake on MMPs, TIMP, and COL-1 Expression in UVB-Irradiated Hairless Mice

Wrinkle formation and skin dehydration are associated with collagen deficiency. The breakdown of collagen, a major skin substance, is regulated by matrix metalloproteinase (MMP) and by the tissue inhibitor of metalloproteinase (TIMP). MMP is a zinc-dependent endopeptidase that is involved in collagen degradation as well as the remodeling of the extracellular matrix (ECM) and is known to play an important role in morphogenesis, skin ulceration, and tumor invasion and metastasis [17,18]. MMP expression is induced by ROS generated upon UV exposure, and after its expression, collagen and elastic fibers (elastin), which are the major components of ECM, are degraded, leading to wrinkle formation and skin dehydration.

Previous studies have shown that MMP-2, MMP-9, and MMP-13 are activated by UVB irradiation and form wrinkles by degrading type IV, type VII, and type I collagen [17,19]. To determine how GOS affects MMPs, the mRNA expression levels of MMP-2, MMP-9, and MMP-13 were measured using PCR (Figure 3). The expression levels of MMP-2, MMP-9, and MMP-13 were lower in the NOR than in the CON (Figure 3A–C). In particular, the expression levels of MMP-9 (*p* < 0.01) and MMP-13 (*p* < 0.001) were significantly lower in the NOR than in the CON. In addition, the GOS and CTP groups showed significantly lower levels of MMP-2 (CTP and GOS: *p* < 0.01), MMP-9 (CTP: *p* < 0.05, GOS: *p* < 0.01), and MMP-13 (CTP and GOS: *p* < 0.001) than those in the CON.

TIMP irreversibly binds to MMPs, specifically inhibits their activity, and is involved in maintaining collagen homeostasis. TIMP-2 and TIMP-1 can be combined non-covalently with MMP-2 and MMP-9 to prevent the direct damage of ECM by MMPs [13]. Decreased expression of TIMP-1 and TIMP-2 breaks the balance between MMP and TIMP, resulting in skin damage and pathological changes caused by MMP [20].

The expression levels of TIMP-1 and TIMP-2 in the NOR were tendentially higher than those in the CON, but no significant difference was observed. However, the expression levels of TIMP-1 (CTP: *p* < 0.05, GOS: *p* < 0.01) and TIMP-2 (CTP and GOS: *p* < 0.001) in the CTP and GOS groups were significantly higher than those in the CON (Figure 3D,E).

ECM components are closely related to the mechanical properties of skin, and the stable structure and arrangement of ECM components imparts skin elasticity. Collagen accounts for 75% of the skin protein and is a vital indicator of skin aging because it plays a role in maintaining skin elasticity and moisture [21]. Among the different types of collagen, COL-1 is primarily associated with elasticity. The expression of COL-1 in the GOS group was significantly higher than that in the CON (Figure 3F, *p* < 0.05).

MMP breaks down the ECM and reduces the content of COL-1. This change is a clear sign of skin damage caused by UV radiation [22]. In this study, GOS oral intake was shown to reduce MMP expression, increase TIMP and COL-1 expression, and protect against skin damage.

### 2.4. Effect of GOS Intake on Cytokine Levels in UVB-Irradiated Hairless Mice

IL-6, IL-12, and TNF-α levels determined by ELISA analysis in the CON group were found to be high compared to those in the normal group (Figure 4: IL-6, 2.28-fold; IL-12, 1.99-fold; TNF-α, 1.97-fold). We also found that the administration of GOS and CTP was associated with a decrease in cytokine levels in skin tissues. GOS and CTP orally-administered groups showed lower IL-6 (CTP and GOS: *p* < 0.001), IL-12 (CTP and GOS: *p* < 0.001), and TNF-α (CTP and GOS: *p* < 0.001) levels than the CON group. GOS and CTP oral intake inhibited the increase in cytokine expression induced by UV irradiation. Ultraviolet irradiation induces the expression of various inflammatory mediators, such as IL-6, IL-8, IL-10, and IL-12, and is directly or indirectly involved in skin inflammatory reactions [23]. These inflammatory cytokines have been reported to stimulate epidermal keratinocytes and skin fibroblasts to regulate MMP levels and break down skin collagen and elastic fibers, thus inducing wrinkles [17].

In our study, UVB irradiation also increased inflammatory cytokines levels significantly. This increase in inflammatory cytokines was less marked upon administration of GOS (Figure 4). According to a study by Tanabe and Hochi [10], oral ingestion of GOS inhibits the production of inflammatory cytokines such as IL-1β, IL-6, IL-17 and tumor necrosis factor in Nc/Nga mice. In addition, by suppressing the production of cytokines associated with skin inflammation, it was found to effectively block atopic dermatitis-like skin lesions in mice. Previous studies have shown that GOS consumption promotes the growth of beneficial bacteria such as *Bifidobacterium* and *Lactobacillus* [24,25]. *Bifidobacterium* and *Lactobacillus* have been reported to enhance the immune function of the host by increasing intestinal immunoglobulin A [26] and enhancing the protective function of the mucosal surface [27]. However, few studies have been conducted on the correlation between increased intestinal beneficial bacteria and decreased inflammatory cytokines. It is necessary to further investigate the relationship between changes in the intestinal flora and inflammatory cytokines.

### 2.5. GOS Intake Prevents MAPK Phosphorylation in UVB-Irradiated Hairless Mice

UV affects inflammatory cells, mainly neutrophils [28]. UV-stimulated neutrophils produce inflammatory cytokines and ROS, causing inflammation through the activation of the mitogen-activated protein kinase (MAPK) pathway [29]. By activating MAPK signaling, the inflammatory response caused by the activation of ERK, JNK, and p38 kinase induces the activation of activator protein-1, increasing the damage to lipids, proteins, nucleic acids, and enzymes and causing skin aging [30].

UVB radiation produces ROS that cause oxidative stress, leading to the activation of MAPK. MAPK signaling molecules were analyzed by Western blotting to investigate the effect of GOS intake on UVB-stimulated MAPK signaling activation. Overexpression of the phosphorylated proteins, p-ERK1 (*p* < 0.001), p-JNK (*p* < 0.001), and p-p38 (*p* < 0.001), was observed in the CON group (Figure 5). The GOS group produced low amounts of p-ERK1, p-JNK, and p-p38. The photoprotective effect of GOS ingestion may be linked to the inhibition of protein production in the MAPK pathway.

UVB irradiation has been shown to stimulate ROS generation and activate the MAPK signaling pathway [31], including ERK1/2, JNK and p38 MAPK [32]. In this study, GOS intake inhibited the UVB-induced phosphorylation of JNK, p38, and ERK. In conclusion, our results suggest that oral administration of GOS prevents skin aging by influencing important gene, cytokine and protein levels associated with wrinkly skin. The inhibitory effects of GOS on skin damage by UVB irradiation includes the reduction in wrinkles through the promotion of water holding capacity and increases inhibition of TEWL. These effects are likely owing to a decrease in the levels of MAPK (i.e., ERK1/2, JNK, and p38).

## 3. Materials and Methods

### 3.1. Materials and Animals

The low molecular weight collagen hydrolysates containing collagen tripeptides (CTP) and GOS (the degree of polymerization (DP) ≥ 2: 21.00 ± 5.66; DP ≥ 3: 31.00 ± 7.07; DP ≥ 4: 20.50 ± 6.36; the linkage type: 1→4 and 1→6) used in this experiment were donated by Neo Crema Co., Ltd (Seoul, Korea).

Six-week-old male albino and hairless mice (SkH: HR-1) were purchased from Orient Bio (Seongnam, Korea) and were given free access to food and water after acclimatization for 1 week. The mice were kept at 22 ± 2 °C with 50 ± 10% relative humidity under a 12 h day/night cycle. All experiments were conducted with the approval of the Institutional Animal Care Use Committee of Korea University (KUIACUC-2019-0062). The mice were randomly divided into four groups of six animals each: normal (NOR; unirradiated mice), ultraviolet (UV) B control (CON; irradiated and untreated mice), CTP intake at 200 mg/kg (CTP; irradiated and CTP-treated mice), and GOS intake at 200 mg/kg (GOS; irradiated and GOS-treated mice).

### 3.2. UVB Exposure to Hairless Mice and Measurement of Skin Parameters

The UV irradiation equipment (BLX-E254, Vilber Lourmat, Marne La Vallee, France) was set to 312 nm UVB wavelength and 790 μW/cm^2^ intensity. Dosimetric monitoring was performed using a Lutron UV-340 light meter (Conrad Electronic AG, Hirschau, Germany). The mice were placed in a UV irradiation cage, and their dorsal skin was exposed to UVB light three times a week for 8 weeks [week 1: 1 MED (Minimal Erythema Dose, 75 mJ/cm^2^); week 2: 2 MED; week 3: 3 MED; weeks 4–8: 4 MED].

TEWL and skin moisture levels were measured using a Tewameter TM300, Corneometer CM825, and Maxameter MX18 (Courage & Khazaka, Köln, Germany), respectively. Skin surface molds were taken using a Visolline^®^ Replica Full Kit (CK Electronic GmbH, Germany) to make replicas and skin wrinkle parameters such as total wrinkle area (mm^2^) and mean wrinkle length (μm) were obtained using a Visoline^®^ VL 650 skin wrinkle analyzer (CK Electronic GmbH).

### 3.3. Gene Expression of Matrix Metalloproteinase and Collagen in Skin Tissue

Total RNA was isolated from skin samples using Trizol^®^ reagent (Invitrogen, Carlsbad, CA, USA), according to the manufacturer’s protocol. First, 1 μg of total RNA was treated with Q1 RNase-free DNase I (Promega, Madison, WI, USA), and complementary DNA (cDNA) was synthesized with 1 μg of total RNA using SuperScript^®^ III reverse transcriptase (Invitrogen). The synthesized cDNA was subjected to real-time PCR using a Power SYBR Green PCR MasterMix kit (Applied Biosystems, Foster City, CA, USA). Data were analyzed using the comparative *C_T_* (2*^−∆∆C^_T_*) method [33], and the results were determined based on a validated control gene, glyceraldehyde-3-phosphate dehydrogenase (*GAPDH*). The following primer sequences were used for real-time PCR: matrix metalloproteinase (MMP) 2 (NM_008610.3), forward primer (FP) 5′-CAAGGATGGACTCCTGGC ACAT-3′, reverse primer (RP) 5′-TACTCGCCATCAGCGTTCCCAT-3′; MMP 9 (NM_013599.4), FP 5′-GCTGACTACGATAAGGACGGCA-3′, RP 5′-TAGTGGTGCAGGCAGAGTAGGA-3′; MMP 13 (NM_008607.2), FP 5′-GATGACCTGTCTGAGGAAGACC-3′, RP 5′-CATTTCTCGGAGCCTGTCAAC-3′; tissue inhibitor of MMP (TIMP) 1 (NM_011593.2), FP 5′-TCTTGGTTCCCTGGCGTACTCT-3′, RP 5′-GTGAGTGTCACTCTCCAGTTTGC-3′; TIMP 2 (NM_011594.3), FP 5′-GCCAAAGCAGTGAGCGAGAAG-3′, RP 5′-GCCGTGTAGATAAACTCGATGTC-3′; type I collagen (COL-1) (NM_007742.4), FP 5′-CCTCAGGGTATTGCTGGACAAC-3′, RP 5′-CAGAAGGACCTTGTTTGCCAGG-3′; and GAPDH (NM_008084.3), FP 5′-CATCACTGCCACCCAGAAGACTG-3′, RP 5′-ATGCCAGTGAGCTTCCCGTTCAG-3′.

### 3.4. Cytokine Assay for Skin Tissue

For the cytokine analysis of skin tissues, 4 × phosphate-buffered saline (pH 7.4) was added to the tissue fragments on ice, and the skin tissue fragments were then ground with a homogenizer (Brinkman Kinematica, Switzerland) for 3 min. The suspension was centrifuged at 3000× *g* and 4 °C for 5 min to recover the supernatant. The protein content of the homogenate was measured using the Bicinchoninic Acid Protein (BCA) Assay Kit (Thermo Scientific Fisher, Rockford, IL, USA). To measure cytokine levels, mouse IL-6, IL-12, and TNF-α ELISA kits were used according to the manufacturer’s protocol (BD Bioscience, San Jose, CA, USA). IL-6, IL-12, and TNF-α levels were expressed as pg/mg protein.

### 3.5. Western Blot Analysis

Skin tissue was first homogenized using a lysis buffer (50 mM Tris-Cl, pH 8.0, 0.1% SDS, 150 mM NaCl, 1% NP-40, 0.02% sodium azide, 0.5% sodium deoxycholate, 100 pg/mL phenylmethylsulfonyl fluoride (PMSF), 1 pg/mL aprotinin, and a phosphatase inhibitor) and a Tissue Lyser II (Qiagen, Venlo, The Netherlands). The lysate was centrifuged at 8000× *g* for 20 min, and the supernatant was used as a protein solution. The protein concentration was measured using the BCA assay kit (Thermo Scientific Fisher), with bovine serum albumin as the standard. Skin lysates containing the same amount of protein (30 μg) were electrophoresed through SDS-PAGE and transferred to PVDF membranes for immunoblotting. Protein expression was detected using specific antibodies and a ChemiDoc^TM^ imaging system (Bio-Rad, Hercules, CA, USA), and the results were normalized to a validated control protein, GAPDH. Primary antibodies were purchased from Abcam (Cambridge, UK), Santa Cruz Biotechnology (Santa Cruz, CA, USA), and Cell Signaling Technology (Danvers, MA, USA); the proteins targeted in Western blotting were extracellular signal regulated kinase (ERK), ERK-phosphorylated (p-ERK), c-Jun N-terminal kinases (JNK), JNK-phosphorylated (p-JNK), p38, and p38-phosphorylated (p-p38).

### 3.6. Statistical Analysis

The experimental results were analyzed using the SPSS (Statistical Package for the Social Science, SPSS Inc., Chicago, IL, USA) version 18.0 program. All results were expressed as mean ± standard deviation (SD). The significance of the results was confirmed by one-way ANOVA, followed by Tukey’s multiple range test at *p* < 0.05.

## Figures and Tables

**Figure 1 molecules-25-01679-f001:**
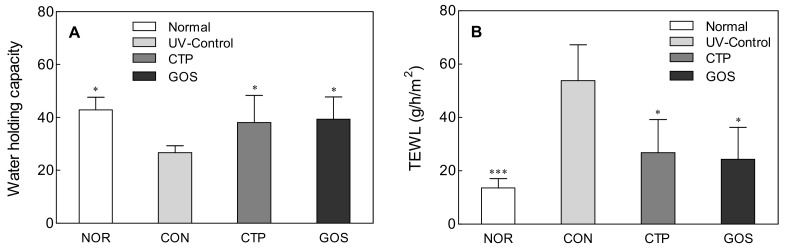
Effect of galacto-oligosaccharide (GOS) intake on the water holding capacity (**A**) and transepidermal water loss (TEWL) (**B**) in ultraviolet (UV) B-irradiated hairless mice. Data are expressed as mean ± standard deviation, and different symbols indicate significant differences at * *p* < 0.05 and *** *p* < 0.001 vs. CON group. Normal (NOR; unirradiated mice); ultraviolet (UV) B control (CON; irradiated and untreated mice); collagen tripeptide (CTP) intake at 200 mg/kg (CTP; irradiated and CTP-treated mice); GOS intake at 200 mg/kg (GOS; irradiated and GOS-treated mice).

**Figure 2 molecules-25-01679-f002:**
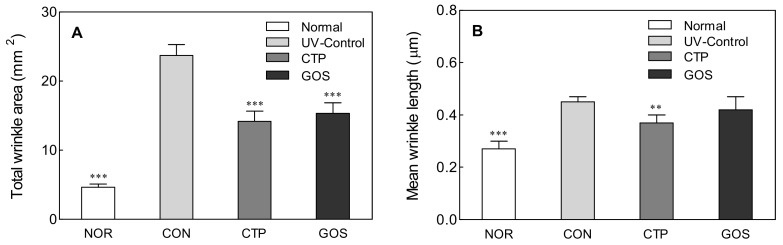
Effect of galacto-oligosaccharide (GOS) intake on total wrinkle area (**A**) and mean wrinkle length (**B**) in ultraviolet (UV) B-irradiated hairless mice. Data are expressed as mean ± standard deviation, and different symbols indicate significant differences at ** *p* < 0.01 and *** *p* < 0.001 vs. CON group. Normal (NOR; unirradiated mice); ultraviolet (UV) B control (CON; irradiated and untreated mice); CTP intake at 200 mg/kg (CTP; irradiated and CTP-treated mice); GOS intake at 200 mg/kg (GOS; irradiated and GOS-treated mice).

**Figure 3 molecules-25-01679-f003:**
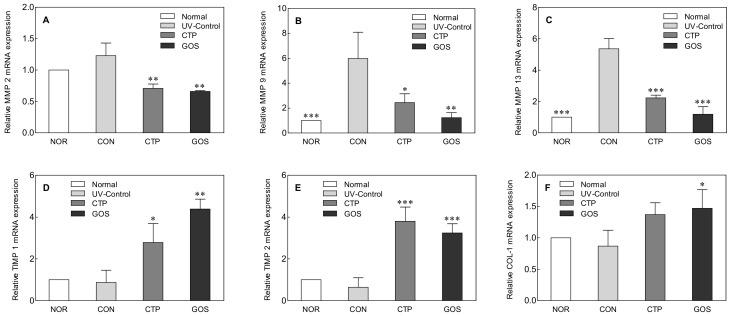
Effect of galacto-oligosaccharide (GOS) intake on matrix metalloproteinase (MMP) (**A**–**C**), tissue inhibitors of MMP (TIMP) (**D**,**E**), and collagen-1 (COL-1) (**F**) mRNA expression in ultraviolet (UV) B-irradiated hairless mice. Data are expressed as mean ± standard deviation, and different symbols indicate significant differences at * *p* < 0.05, ** *p* <0.01 and *** *p* < 0.001 vs. CON group. Normal (NOR; unirradiated mice); ultraviolet (UV) B control (CON; irradiated and untreated mice); CTP intake at 200 mg/kg (CTP; irradiated and CTP-treated mice); GOS intake at 200 mg/kg (GOS; irradiated and GOS-treated mice).

**Figure 4 molecules-25-01679-f004:**
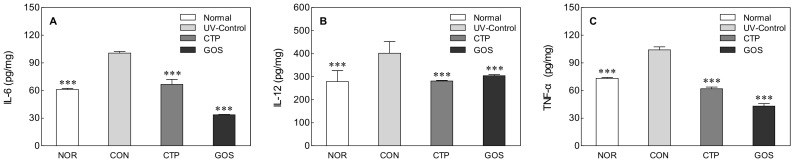
Effect of galacto-oligosaccharide (GOS) intake on interleukin (IL)-6 (**A**), IL-12 (**B**), and tumor necrosis factor-α (TNF-α, **C**) levels in ultraviolet (UV) B-irradiated hairless mice. Data are expressed as mean ± standard deviation, and different symbols indicate significant differences at *** *p* < 0.001 vs. CON group. Normal (NOR; unirradiated mice); ultraviolet (UV) B control (CON; irradiated and untreated mice); CTP intake at 200 mg/kg (CTP; irradiated and CTP-treated mice); GOS intake at 200 mg/kg (GOS; irradiated and GOS-treated mice).

**Figure 5 molecules-25-01679-f005:**
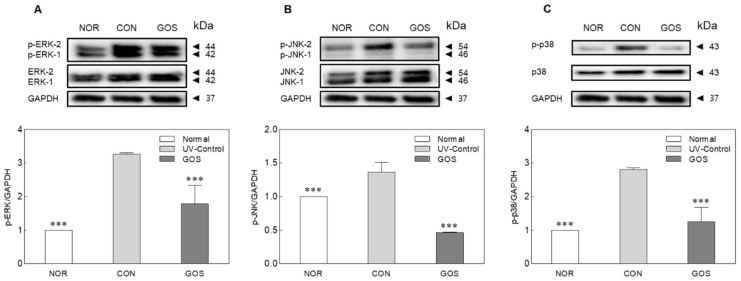
Effect of galacto-oligosaccharide (GOS) intake on the phosphorylated extracellular signaling regulated kinase (ERK, **A**), Jun N-terminal kinase (JNK, **B**) and p38 (**C**) levels in ultraviolet (UV) B-irradiated hairless mice, as determined by Western blotting. Data are expressed as mean ± standard deviation, and different symbols indicate significant differences at *** *p* < 0.001 vs. CON group. Normal (NOR; unirradiated mice); ultraviolet (UV) B control (CON; irradiated and untreated mice); CTP intake at 200 mg/kg (CTP; irradiated and CTP-treated mice); GOS intake at 200 mg/kg (GOS; irradiated and GOS-treated mice); glyceraldehyde 3-phosphate dehydrogenase (GAPDH).

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
