# Peer review of "Photoprotective Effect of Dietary Galacto-Oligosaccharide (GOS) in Hairless Mice via Regulation of the MAPK Signaling Pathway"

_molecules, 2020, doi:10.3390/molecules25071679_

Round 1

Reviewer 1 Report

In the case of English quality, I’m not an expert because is not my native language.

-The manuscript is a research project about the photoprotective properties of galactooligosaccharides, involving robust biological tests on rats feed with this prebiotic, including gene expression of genes involved in UV damage.

In my knowledge, this is the first report of the photoprotective of galactooligosaccharides, I think they measured biological parametrs with robust tests in order to investigate the effect of GOS, including gene expression involved in UV damage, according with results, they proved the photoprotective effect of GOS. The discussion explain also these foundings and a good comparison with the scientific literature was carried out.

It could be interesting to know the linkage type of the GOS mixture in the prebiotic, in order to know or explain structure/function of GOS.

It could be interesting to know in more detail the GOS and CTP composition, in the case of GOS, ß-galactosyl linkage type and DP.

Author Response

First of all, we would like to thank you for your interest upon our work and for the valuable comments that have helped us to improve the quality of our paper.

We have revised our manuscript according to the editor and reviewers’ comments and response to all comments from reviewers point by point.

Point 1: It could be interesting to know the linkage type of the GOS mixture in the prebiotic, in order to know or explain structure/function of GOS.

Response 1: The sentence in the introduction section has been revised and a reference has been added.

Line 50-53: GOS are potent non-digestible prebiotics composed of 3-10 molecules of galactose and glucose, and the elongation products of β-galactosidase. The resulting molecules are known to contain a varying degree of β-glycosidic linkages and promote digestive and immune health through the growth of beneficial bacteria [7].

Line 299-301: 7.  Oh, SY.; Youn, SY.; Park, MS.; Kim, HG.; Baek, NI.; Li, Z.; Ji, GE. Synthesis of β-galactooligosaccharide using Bifidobacterial β-galactosidase purified from recombinant Escherichia coli. J. Microbiol. Biotechnol. 2017, 27, 1392-1400.

Point 2: It could be interesting to know in more detail the GOS and CTP composition, in the case of GOS, ß-galactosyl linkage type and DP.

Response 2: The information of β-galactosyl linkage type and DP have been added in the materials and methods section.

Line 203-206: The low molecular weight collagen hydrolysates containing collagen tripeptides (CTP) and GOS [the degree of polymerization (DP) ≥ 2: 21.00 ± 5.66; DP ≥ 3: 31.00 ± 7.07; DP ≥ 4: 20.50 ± 6.36; the linkage type: 1→4 and 1→6]  used in this experiment were donated by Neo Crema Co. Ltd (Seoul, Republic of Korea).

Reviewer 2 Report

The manuscript delieved excited news for photoaging. The potential benefits of  galacto-oligosaccharide would attract readers' attention. Only one question I have is the bioavailablity of GSO in mice. How about let mice skin smear with it? 

Author Response

First of all, we would like to thank you for your interest upon our work and for the valuable comments that have helped us to improve the quality of our paper.

We have revised our manuscript according to the editor and reviewers’ comments and response to all comments from reviewers point by point.

  • Response to Reviewer 2 Comments

Point 1: The manuscript delieved excited news for photoaging. The potential benefits of  galacto-oligosaccharide would attract readers' attention. Only one question I have is the bioavailablity of GOS in mice. How about let mice skin smear with it?

Response 1: We evaluated the potential of GOS as a cosmetic material by measuring the changes of skin factors and skin microorganisms upon applying serum cosmetics containing GOS, a prebiotic, to the subjects. According to the analysis of the results, applying GOS to cosmetics will have a beneficial effect on the control of skin microbes. Currently, we are preparing to submit a manuscript using these clinical trial results.

Reviewer 3 Report

Manuscript: Molecules-736008, titled “Photoprotective Effect of Dietary galacto-oligosaccharide (GOS) in Hairless Mice via Regulation of the MAPK Signaling Pathway.”, by Min Guen Suh et al.

The manuscript Molecules-736008 aims to provide a molecular basis for the reported effect of dietary supplementation with the prebiotic Galatto-Oligosaccaride (GOS) on inducing a beneficial modulation of the intestinal microbiota and a protective effect on skin under skin damage and atopic dermatitis.

            To this purpose the authors present data obtained in hairless mice challenged and unchallenged with UV radiation and treated or non-treated with a dietary supplementation with GOS and with collagen tripeptide, here used as a reference protective treatment.

Data about skin wrinkles formation and extension and moisture content are presented together with the modulation of inflammatory cytokines expression and release, MMPs and TIMP expression and MAPK activation. UV-B protection through systemic drug administration is a relevant issue and data are of potential interest to the readers. However, the manuscript suffers of a number of flaws that need to be corrected.

The results are not clearly presented, the text is difficult to follow and to understand.

Throughout the paper and figures, authors refer to unirradiated cells as “Normal” (acronym “NOR”) and to irradiated; GOS or CTP untreated cells as Control (acronym “CON”). These definitions are misleading and should be changed to a more rigorously descriptive term as, for instance, “control unirradiated cells “, “irradiated, untreated cells”. These definitions first appear in figure 1 (line 82) but their corresponding acronyms are introduced just in the section “matierials”, at line 197!

Throughout the paper plots report data and SEM instead of SD. A reader wants to know how much experimental values are scattered apart (=Standard Deviation) and not how distant is the mean of experimental data from the mean of the true data(=SEM) (See, Intuitive Biostatistics, A Nonmathematical Guide to Statistical Thinking, Fourth Edition. Harvey Motulsky, Oxford Univ. Press - November 2017).  

On each histogram a different letter is reported (a,b, c, etc) and the legend states “and the different letters indicate significant differences (p < 0.05)”.

What does it mean? Do histograms have different p values? If yes, what these different values are? If no, and the histograms have the same P<0.05 value, why they have different letters?

The authors report about a reduction in Wrinkles’ number and extension, but no method, nor reference is described for their evaluation.

References are used, in scientific manuscripts, to acknowledge precedent achievements and to support undemonstrated statements, usually through citing the first experimental demonstration or a recent authoritative review.

In this paper a number of references are inappropriately quoted and discussed. E.g.:  at line 125-127, to stand the sentence that “Decreased expression of TIMP-1 and TIMP-2  breaks the balance between MMP and TIMP, resulting in skin damage and even pathological changes caused by MMP”,  the reference [19]  Devi S,  et al 2017, is cited. However in this reference the TIMP-1 and TIMP-2 are never mentioned, while the paper is focused on some possible antioxidant properties associated with Buffalo casein.

Other inappropriately cited references are ref no 20, 21, 26 and 27.

A complete linguistic revision is needed.v

Author Response

First of all, we would like to thank you for your interest upon our work and for the valuable comments that have helped us to improve the quality of our paper.

We have revised our manuscript according to the editor and reviewers’ comments and response to all comments from reviewers point by point.

  • Response to Reviewer 3 Comments

Point 1: Throughout the paper and figures, authors refer to unirradiated cells as “Normal” (acronym “NOR”) and to irradiated; GOS or CTP untreated cells as Control (acronym “CON”). These definitions are misleading and should be changed to a more rigorously descriptive term as, for instance, “control unirradiated cells “, “irradiated, untreated cells”. These definitions first appear in figure 1 (line 82) but their corresponding acronyms are introduced just in the section “materials”, at line 197!

Response 1: The sentence in the materials and methods section has been revised. In addition, a description of the acronym has been added to the figure legend.

Line 211-214: normal (NOR; unirradiated mice), ultraviolet (UV) B control (CON; irradiated and untreated mice), CTP intake at 200 mg/kg (CTP; irradiated and CTP treated mice), and GOS intake at 200 mg/kg (GOS; irradiated and GOS treated mice).

Line 86-88, 103-105, 127-129, 161-164, 190-192: Normal (NOR; unirradiated mice); ultraviolet (UV) B control (CON; irradiated and untreated mice); CTP intake at 200 mg/kg (CTP; irradiated and CTP treated mice); GOS intake at 200 mg/kg (GOS; irradiated and GOS treated mice).

Point 2: Throughout the paper plots report data and SEM instead of SD. A reader wants to know how much experimental values are scattered apart (=Standard Deviation) and not how distant is the mean of experimental data from the mean of the true data(=SEM) (See, Intuitive Biostatistics, A Nonmathematical Guide to Statistical Thinking, Fourth Edition. Harvey Motulsky, Oxford Univ. Press - November 2017). 

Response 2: We revised the error bar for all figures to the standard deviation and the sentence in the materials and methods and figure legend section has been revised.

Line 272-273: All results were expressed as mean ± standard deviation (SD).

Line 84-85, 102, 126, 160-161, 189: Data are expressed as means ± standard deviation

Point 3: On each histogram a different letter is reported (a,b, c, etc) and the legend states “and the different letters indicate significant differences (p < 0.05)”. What does it mean? Do histograms have different p values? If yes, what these different values are? If no, and the histograms have the same P<0.05 value, why they have different letters?

Response 3: We have performed ANOVA followed by Turkey’s Multiple comparison in SPSS software. The different letters (a, b, c) in the figure all show significant differences at the same level of probability (p <0.05). This means that the probability of a and b being different is more than 95%.

Point 4: The authors report about a reduction in Wrinkles’ number and extension, but no method, nor reference is described for their evaluation. References are used, in scientific manuscripts, to acknowledge precedent achievements and to support undemonstrated statements, usually through citing the first experimental demonstration or a recent authoritative review.

Response 4: The method of wrinkle measurement has been added in the materials and methods part.

Line 223-226: Skin surface molds were taken using a Visolline® Replica Full Kit (CK Electronic GmbH, Germany) to make replicas and skin wrinkle parameters such as total wrinkle area (mm2) and mean wrinkle length (μm) were obtained using a Visoline® VL 650 skin wrinkle analyzer (CK Electronic GmbH). 

Point 5: In this paper a number of references are inappropriately quoted and discussed. E.g.:  at line 125-127, to stand the sentence that “Decreased expression of TIMP-1 and TIMP-2  breaks the balance between MMP and TIMP, resulting in skin damage and even pathological changes caused by MMP”,  the reference [19]  Devi S,  et al 2017, is cited. However, in this reference the TIMP-1 and TIMP-2 are never mentioned, while the paper is focused on some possible antioxidant properties associated with Buffalo casein.

Response 5: We replaced the correct reference in the sentence pointed out by the reviewer.

Line 328-329: 20. Bellayr, IH.; Mu, X.; Li Y.  Biochemical insights into the role of matrix metalloproteinases in regeneration: challenges and recent developments. Future. Med. Chem. 2009, 1, 1095-1111.

Point 6: Other inappropriately cited references are ref no 20, 21, 26 and 27.

Response 6: We replaced the correct references in the discussion and reference sections.

Line 330: 21. Shoulders, MD.; Raines, RT. Collagen structure and stability. Annu. Rev. Biochem. 2009, 78, 929-958.

Line 331-333: 22. Quan, T.; Little, E.; Quan, H.; Qin, Z.; Voorhees, JJ.; Fisher, GJ. Elevated matrix metalloproteinases and collagen fragmentation in photodamaged human skin: impact of altered extracellular matrix microenvironment on dermal fibroblast function. J. Invest. Dermatol. 2013, 133, 1362-1366.

Line 344-345: 27. Scharffetter-Kochanek, K.; Wlaschek, M.; Brenneisen, P.; Schauen, M.; Blaudschun, R.; Wenk, J. UV-induced reactive oxygen species in photocarcinogenesis and photoaging. Biol. Chem. 1997, 378, 1247–1257.

Line 346-347: 28. Zhang, J.; Wang, X.; Vikash, V.; Ye, Q.; Wu, D.; Liu, Y.; Dong, W. ROS and ROS-mediated cellular signalling. Oxid. Med. Cell. Longev. 2016. 2016, 1-18.

Point 7: A complete linguistic revision is needed.

Response 7: Our marked manuscript was reedited by editing service of Editage.

Round 2

Reviewer 3 Report

Dear editor,

the authors did a great effort to improve their work, nonetheless a few critical points remain:

1) as formerly highlighted, the acronyms “CON” and “NOR” must be clearly defined when they first appear in the main text, i.e. at lines 76 and 79 and not in the figure legends or in materials and methods (at the end of the manuscript).

2) Again, as formerly requested, in each figure, histograms are headed by a different letter (a, b, c, d) and the legend states “… different letters indicate significant differences (p < 0.05)”. Authors must openly state the p value each letter is intended to represent.  

In addition, in figure 3 panel F, a 1.5-fold increase, with a conspicuous SD overlapping the one of CON sample, is depicted for Col-1RNA. Such a variation has clearly no biological meaning and the comment at lines 142-143: “The expression of COL-1 in the CTP and GOS groups was significantly higher than that in the CON (Fig. 3F)” is misleading and must be changed to account the real state of things.

At lines 166-172, to comment fig 4, the authors propose the gut-brain-skin axis theory as a fact instead as a controversial hypothesis, and the chain of events linking prebiotics intake, gut microbiome modification, Immune system response, inflammation reduction and UV protection as an experimentally proved mechanisms while providing no direct experimental evidence for them.

This passage must be re-written to clearly account the merely speculative nature of this chain of events.

At lines 196-197, the conclusion that “…GOS intake suppresses skin damage caused by UV light” Is a coarse overstament! Instead, presented data are indicate that a GOS dietary supplementation is able to partially revert/ counteract a few biochemical effects induced by UV radiation. The conclusions have to be changed to strictly reflect the actual experimental results.

Author Response

First of all, we would like to thank you for your interest upon our work and for the valuable comments that have helped us to improve the quality of our paper.

We have revised our manuscript according to the editor and reviewers’ comments and response to all comments from reviewers point by point.

  • Response to Reviewer 3 Comments

Point 1: as formerly highlighted, the acronyms “CON” and “NOR” must be clearly defined when they first appear in the main text, i.e. at lines 76 and 79 and not in the figure legends or in materials and methods (at the end of the manuscript).

Response 1: The sentence in the main text of the results and discussion section has been revised.

Line 76-81: In the ultraviolet (UV) B control (CON; irradiated and untreated mice) group, the water holding capacity and TEWL significantly changed compared to those in the normal (NOR; unirradiated mice, Fig. 1A: p<0.05, Fig. 1B: p<0.001) group. In contrast, the water holding capacity of both GOS intake at 200 mg/kg (GOS; irradiated and GOS treated mice) and collagen tripeptide intake at 200 mg/kg (CTP; irradiated and CTP treated mice) groups was significantly higher than that of the CON (Fig. 1A, CTP and GOS: p<0.05).

Point 2: Again, as formerly requested, in each figure, histograms are headed by a different letter (a, b, c, d) and the legend states “… different letters indicate significant differences (p < 0.05)”. Authors must openly state the p value each letter is intended to represent. 

Response 2: According to the reviewer’s opinion, all parts of the thesis were amended as follows.

Line 76-78: In the ultraviolet (UV) B control (CON; irradiated and untreated mice) group, the water holding capacity and TEWL significantly changed compared to those in the normal (NOR; unirradiated mice, Fig. 1A: p<0.05, Fig. 1B: p<0.001) group.

Line 78-82: In contrast, the water holding capacity of both GOS intake at 200 mg/kg (GOS; irradiated and GOS treated mice) and collagen tripeptide intake at 200 mg/kg (CTP; irradiated and CTP treated mice) groups was significantly higher than that of the CON (Fig. 1A, CTP and GOS: p<0.05). No significant difference was noted in the NOR. TEWL was also significantly lower in the GOS and CTP groups than in the CON (Fig. 1B, CTP and GOS: p<0.05).

Line 88-89: different symbols indicate significant differences at *p<0.05 and ***p<0.001 vs. CON group.

Line 98-102: The wrinkle area and mean wrinkle length significantly increased in the CON compared to those in the NOR (Fig. 2A and 2B: p<0.001). In addition to UVB treatment, treatment with GOS and CTP significantly lowered the wrinkle area (CTP and GOS: p<0.001) and mean wrinkle length (CTP: p<0.01) compared to those in the CON.

Line 107: different symbols indicate significant differences at **p<0.01 and ***p<0.001 vs. CON group.

Line 124-127: In particular, the expression levels of MMP-9 (p<0.01) and MMP-13 (p<0.001) were significantly lower in the NOR than in the CON. In addition, the GOS and CTP groups showed significantly lower levels of MMP-2 (CTP and GOS: p<0.01), MMP-9 (CTP: p<0.05, GOS: p<0.01), and MMP-13 (CTP and GOS: p<0.001) than those in the CON.

Line 131-132: different symbols indicate significant differences at *p<0.05, **p<0.01 and ***p<0.001 vs. CON group.

Line 142-144: However, the expression levels of TIMP-1 (CTP: p<0.05, GOS: p<0.01) and TIMP-2 (CTP and GOS: p<0.001) in the CTP and GOS groups were significantly higher than those in the CON (Fig. 3D, E).

Line 149-150: The expression of COL-1 in the GOS group was significantly higher than that in the CON (Fig. 3F, p<0.05).

Line 158-159: GOS and CTP orally administered groups showed lower IL-6 (CTP and GOS: p<0.001), IL-12 (CTP and GOS: p<0.001), and TNF-a (CTP and GOS: p<0.001) levels than the CON.

Line 168-169: different symbols indicate significant differences at ***p<0.001 vs. CON group.

Line193-194: Overexpression of the phosphorylated proteins, p-ERK1 (p<0.001), p-JNK (p<0.001), and p-p38 (p<0.001), was observed in the CON (Fig. 5).

Line201: different symbols indicate significant differences at ***p<0.001 vs. CON group.

Point 3: In addition, in figure 3 panel F, a 1.5-fold increase, with a conspicuous SD overlapping the one of CON sample, is depicted for Col-1RNA. Such a variation has clearly no biological meaning and the comment at lines 142-143: “The expression of COL-1 in the CTP and GOS groups was significantly higher than that in the CON (Fig. 3F)” is misleading and must be changed to account the real state of things.

Response 3: In the Fig. 3F, the letter for the p value of the CTP group was a typo error. The sentence has been revised.

Line 149-150: The expression of COL-1 in the GOS group was significantly higher than that in the CON (Fig. 3F, p<0.05).

Point 4: At lines 166-172, to comment fig 4, the authors propose the gut-brain-skin axis theory as a fact instead as a controversial hypothesis, and the chain of events linking prebiotics intake, gut microbiome modification, Immune system response, inflammation reduction and UV protection as an experimentally proved mechanisms while providing no direct experimental evidence for them. This passage must be re-written to clearly account the merely speculative nature of this chain of events.

Response 4: The sentence was rewritten and a new reference was added.

Line 173-183: According to a study by Tanabe and Hochi [24], oral ingestion of GOS inhibits the production of inflammatory cytokines such as IL-1β, IL-6, IL-17 and tumor necrosis factor in Nc/Nga mice. In addition, by suppressing the production of cytokines associated with skin inflammation, it was found to effectively block atopic dermatitis-like skin lesions in mice. Previous studies have shown that GOS consumption promotes the growth of beneficial bacteria such as Bifidobacterium and Lactobacillus [25, 26]. Bifidobacterium and Lactobacillus have been reported to enhance the immune function of the host by increasing intestinal immunoglobulin A [27] and enhancing the protective function of the mucosal surface [28]. However, few studies have been conducted on the correlation between increased intestinal beneficial bacteria and decreased inflammatory cytokines. It is necessary to further investigate the relationship between changes in the intestinal flora and inflammatory cytokines.

Line 349-350: 24. Tanabe, S.; Hochi, S. Oral administration of a galactooligosaccharide preparation inhibits development of atopic dermatitis-like skin lesions in NC/Nga mice. Int. J. Mol. Med. 2010, 25, 331-336.

Line 351-354: 25. Roberfroid, M.;, Gibson, G.R.; Hoyles, L.; McCartney, A.L.; Rastall, R.; Rowland, I.; Wolvers, D.; Watzl, B.; Szajewska, H.; Stahl, B.; Guarner, F.; Respondek, F.; Whelan, K.; Coxam, V.; Davicco, M.J.; Léotoing, L.; Wittrant, Y.; Delzenne, N.M.; Cani, P.D.; Neyrinck, A.M.; Meheust, A. Prebiotic effects: metabolic and health benefits. Br. J. Nutr. 2010. 104, 1-63.

Line 355-356: 26. Delzenne, N.M.; Neyrinck, A.M.; Cani, P.D. Gut microbiota and metabolic disorders: How prebiotic can work? Br. J. Nutr. 2013. 109, 81-85.

Line 357-358: 27. Erickson, K.L.; Hubbard, N.E. Probiotic immunomodulation in health and disease. J. Nutr. 2000. 130, 403-409.

Line 359: 28. Dow, W.F. The intestinal immune system. Gut. 1989. 30, 1679-1685.

Point 5: At lines 196-197, the conclusion that “…GOS intake suppresses skin damage caused by UV light” Is a coarse overstament! Instead, presented data are indicate that a GOS dietary supplementation is able to partially revert/ counteract a few biochemical effects induced by UV radiation. The conclusions have to be changed to strictly reflect the actual experimental results.

Response 5: The sentence has been revised.

Line 207-209: In conclusion, GOS intake suppresses skin damage caused by UV light our results suggest that oral administration of GOS prevents skin aging by influencing important gene, cytokine and protein levels associated with wrinkly skin.
